# How to Avoid Being Eaten by a Grue: Structured Exploration Strategies for Textual Worlds

## Abstract

Text-based games are long puzzles or quests, characterized by a sequence of sparse and potentially deceptive rewards. They provide an ideal platform to develop agents that *perceive* and *act upon* the world using a combinatorially sized natural language state-action space. Standard Reinforcement Learning agents are poorly equipped to effectively explore such spaces and often struggle to overcome bottlenecks—states that agents are unable to pass through simply because they do not see the right action sequence enough times to be sufficiently reinforced. We introduce Q*BERT, an agent that learns to build a knowledge graph of the world by answering questions, which leads to greater sample efficiency. To overcome bottlenecks, we further introduce MC!Q*BERT an agent that uses an knowledge-graph-based intrinsic motivation to detect bottlenecks and a novel exploration strategy to efficiently learn a chain of policy modules to overcome them. We present an ablation study and results demonstrating how our method outperforms the current state-of-the-art on nine text games, including the popular game, *Zork*, where, for the first time, a learning agent gets past the bottleneck where the player is eaten by a Grue.

## 1 Introduction

Text-adventure games such as *Zork1* (Anderson et al., 1979) (Fig. 1) are simulations featuring language-based state and action spaces. Prior game playing works have focused on a few challenges that are inherent to this medium: (1) *Partial observability* the agent must reason about the world solely through incomplete textual descriptions (Narasimhan et al., 2015; Côté et al., 2018; Ammanabrolu & Riedl, 2019b). (2) *Commonsense reasoning* to enable the agent to more intelligently interact with objects in its surroundings (Fulda et al., 2017; Yin & May, 2019; Adolphs & Hofmann, 2019; Ammanabrolu & Riedl, 2019a). (3) *A combinatorial state-action space* wherein most games have action spaces exceeding a billion possible actions per step; for example the game *Zork1* has $1.64 \times 10^{14}$ possible actions at every step (Hausknecht et al., 2020; Ammanabrolu & Hausknecht, 2020). Despite these challenges, modern text-adventure agents such as KG-A2C (Ammanabrolu & Hausknecht, 2020), TDQN (Hausknecht et al., 2020), and DRRN (He et al., 2016) have relied on surprisingly simple exploration strategies such as $\epsilon$-greedy or sampling from the distribution of possible actions.

Most text-adventure games have relatively linear plots in which players must solve a sequence of puzzles to advance the story and gain score. To solve these puzzles, players have freedom to a explore both new areas and previously unlocked areas of the game, collect clues, and acquire tools needed to solve the next puzzle and unlock the next portion of the game. From a Reinforcement Learning perspective, these puzzles can be viewed as bottlenecks that act as partitions between different regions of the state space. We contend that existing Reinforcement Learning agents that are unaware of such latent structure and are thus poorly equipped for solving these types of problems.

In this paper we introduce two new agents: Q*BERT and MC!Q*BERT, both designed with this latent structure in mind. The first agent, Q*BERT, improves on existing text-game agents that use knowledge graph-based state representations by framing knowledge graph construction during exploration as a question-answering task. To train Q*BERT's knowledge graph extractor, we introduce the *Jericho-QA* dataset for question-answering in text-games. We show that it leads to improved knowledge graph

accuracy and sample efficiency compared to prior methods for constructing knowledge graphs in text-games (Ammanabrolu & Riedl, 2019b).

However, improved knowledge graph accuracy is not enough to overcome bottlenecks; it does not improve asymptotic performance. To this end, MC!Q*BERT (Modular policy Chaining! Q*BERT) extends Q*BERT by combining two innovations: (1) an intrinsic motivation based on expansion of its knowledge graph both as a way to encourage exploration as well as a means for the agent to self-detect when it is stuck; and (2) by additionally introducing a structured exploration algorithm that, when stuck on a bottleneck, will backtrack through the sequence of states leading to the current bottleneck, in search of alternative solutions. As MC!Q*BERT overcomes bottlenecks, it constructs a modular policy that chains together the solutions to multiple bottlenecks. Like Go Explore (Ecoffet et al., 2019), MC!Q*BERT relies on the determinism present in many text-games to reliably revisit previous states. However, we show that

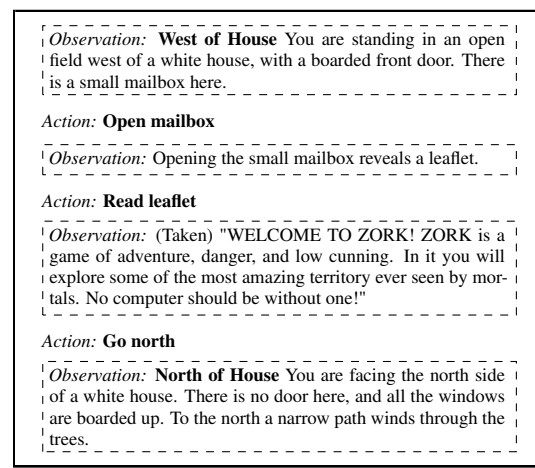

| |
|---|
| *Observation:* **West of House** You are standing in an open field west of a white house, with a boarded front door. There is a small mailbox here. |
| *Action:* **Open mailbox** |
| *Observation:* Opening the small mailbox reveals a leaflet. |
| *Action:* **Read leaflet** |
| *Observation:* (Taken) "WELCOME TO ZORK! ZORK is a game of adventure, danger, and low cunning. In it you will explore some of the most amazing territory ever seen by mortals. No computer should be without one!" |
| *Action:* **Go north** |
| *Observation:* **North of House** You are facing the north side of a white house. There is no door here, and all the windows are boarded up. To the north a narrow path winds through the trees. |

Figure 1: Excerpt from *Zork1*.

MC!Q*BERT's ability to detect bottlenecks via the knowledge graph state representation enable it to outperform such alternate exploration strategies on nine different games.

Our contributions are as follows: 1) We develop an improved knowledge-graph extraction procedure based on question answering and introduce the open-source *Jericho-QA* training dataset. 2) We show that intrinsic motivation reward based on knowledge graph expansion is capable of reliably identifying bottleneck states. 3) Finally, we show that structured exploration in the form of backtracking can be used to overcome these bottleneck states and reach state-of-the-art levels of performance on the Jericho benchmark (Hausknecht et al., 2020).

## 2 UNDERSTANDING BOTTLENECK STATES

Overcoming bottlenecks is not as simple as selecting the correct action from the bottleneck state. Most bottlenecks have long-range dependencies that must first be satisfied: *Zork1* for instance features a bottleneck in which the agent must pass through the unlit *Cellar* where a monster known as a Grue lurks, ready to eat unsuspecting players who enter without a light source. To pass this bottleneck the player must have previously acquired and lit the latern. Other bottlenecks don't rely on inventory items and instead require the player to have satisfied an external condition such as visiting the reservoir control to drain water from a submerged room before being able to visit it. In both cases, the actions that fulfill dependencies of the bottleneck, e.g. acquiring the lantern or draining the room, are not rewarded by the game. Thus agents must correctly satisfy all *latent* dependencies, most of which are unrewarded, then take the right action from the correct location to overcome such bottlenecks. Consequently, most existing agents—regardless of whether they use a reduced action space (Zahavy et al., 2018; Yin & May, 2019) or the full space (Hausknecht et al., 2020; Ammanabrolu & Hausknecht, 2020)—have failed to consistently clear these bottlenecks.

To better understand how to design algorithms that pass these bottlenecks, we first need to gain a sense for what they are. We observe that quests in text games can be modeled in the form of a dependency graph. These dependency graphs are directed acyclic graphs (DAGs) where the vertices indicate either rewards that can be collected or dependencies that must be met to progress and are generally unknown to a player *a priori*. In text-adventure games the dependencies are of two types: items that must be collected for future use, and locations that must be visited. An example of such a graph for the game of *Zork1* can found in Fig. 2. More formally, bottleneck states are vertices in the dependency graph that, when the graph is laid out topographically, are (a) the only state on a level, and (b) there is another state at a higher level with non-zero reward. Bottlenecks can be mathematically expressed as follows: let $\mathcal{D} = \langle V, E \rangle$ be the directed acyclic dependency graph for a particular game

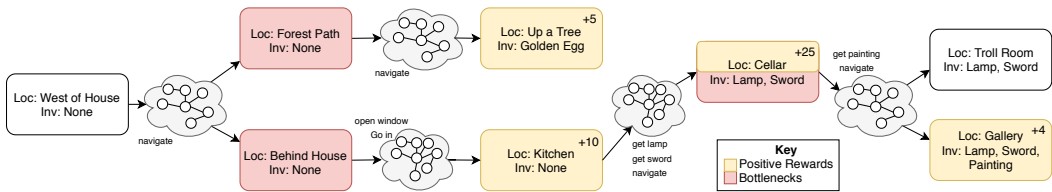

Figure 2: Portion of the *Zork1* quest structure visualized as a directed acyclic graph. Each node represents a state; clouds represent areas of high branching factor with labels indicating some of the actions that must be performed to progress

where each vertex is tuple $v = \langle s_l, s_i, r(s) \rangle$ containing information on some state $s$ such that $s_l$ are location dependencies, $s_i$ are inventory dependencies, and $r(s)$ is the reward associated with the state. There is a directed edge $e \in E$ between any two vertices such that the originating state meets the requirements $s_l$ and $s_i$ of the terminating vertex. $\mathcal{D}$ can be topologically sorted into levels $L = \{l_1, ..., l_n\}$ where each level represents a set of game states that are not dependant on each other. We formulate the set of all bottleneck states in the game:

$$\mathcal{B} = \{b : (|l_i| = 1, b \in l_i, V) \wedge (\exists s \in l_j \text{ s.t. } (j > i \wedge r(s) \neq 0))\} \tag{1}$$

This reads as the set of all states that that belong to a level with only one vertex and that there exists some state with a non-zero reward that depends on it. Intuitively, regardless of the path taken to get to a bottleneck state, any agent must pass it in order to continue collecting future rewards. *Behind House* is an example of a bottleneck state as seen in Fig. 2. The branching factor before and after this state is high but it is the only state through which one can enter the *Kitchen* through the window.

## 3 RELATED WORK AND BACKGROUND

We use the definition of text-adventure game as Partially-Observable Markov Decision Process (Côté et al., 2018; Hausknecht et al., 2020). A game can be represented as a 7-tuple of $\langle S, T, A, \Omega, O, R, \gamma \rangle$ representing the set of environment states, *mostly deterministic conditional transition probabilities between states*, the vocabulary or words used to compose text commands, observations returned by the game, observation conditional probabilities, reward function, and the discount factor respectively. LSTM-DQN (Narasimhan et al., 2015) and Action Elimination DQN (Zahavy et al., 2018) operate on a reduced action space of the order of $10^2$ actions per step by considering either verb-noun pairs or by using a walkthrough of the game respectively. The agents learn how to produce Q-value estimates that maximize long term expected reward. The DRRN algorithm for choice-based games (He et al., 2016; Zelinka, 2018) estimates Q-values for a particular action from a particular state. Fulda et al. (2017) use word embeddings to model affordances for items in these games.

**Exploration strategies:** Jain et al. (2019) extend consistent Q-learning (Bellemare et al., 2016) to text-games, focusing on taking into account historical context. In terms of exploration strategies, Yuan et al. (2018) detail how counting the number of unique states visited improves generalization in unseen games. Yuan et al. (2019) introduce the concept of interactive question-answering in the form of *QAit*—modeling QA tasks in *TextWorld* (Côté et al., 2018), a simplified text-game testbed.

**Knowledge Graphs:** Ammanabrolu & Riedl (2019b) introduce KG-DQN, using knowledge graphs as state representation for text-game agents; Ammanabrolu & Riedl (2019a); Murugesan et al. (2020) explore transfer of commonsense knowledge in text-games with knowledge graphs. Ammanabrolu & Hausknecht (2020) showcase the KG-A2C, for the first time tackling the fully combinatorial action space and presenting state-of-the-art results on many man-made text games. In a similar vein, Adhikari et al. (2020) present the Graph-Aided Transformer Agent (GATA) which learns to construct a knowledge graph during game play and improves zero-shot generalization on procedurally generated *TextWorld* (Côté et al., 2018) games.

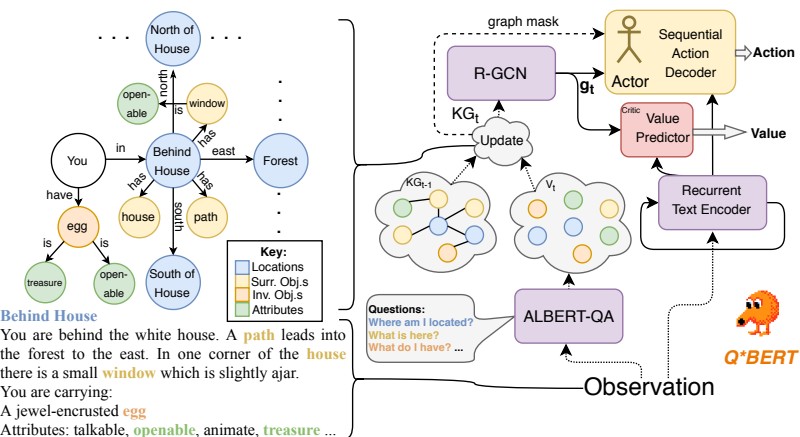

Figure 3: One-step knowledge graph extraction in the Jericho-QA format, and overall Q*BERT architecture at time step $t$. At each step the ALBERT-QA model extracts a relevant highlighted entity set $V_t$ by answering questions based on the observation, which is used to update the knowledge graph.

## 4   Q*BERT

Q*BERT is a reinforcement learning agent that uses a knowledge-graph to represent its understanding of the world state. A knowledge graph (Fig. 3) is a set of relations $\langle s, r, o \rangle$ such that $s$ is a subject, $r$ is a relation, and $o$ is an object. Instead of using relation extraction rules as in KG-A2C (Ammanabrolu & Hausknecht, 2020), Q*BERT uses a variant of the BERT (Devlin et al., 2018) natural language transformer to answer questions and populate the knowledge graph from the answers.

**Knowledge Graph State Representation**   We treat the problem of constructing the knowledge graph as a question-answering task. Our method first extracts a set of graph vertices $\mathcal{V}$ by asking a question-answering system relevant questions and then linking them together using a set of relations $\mathcal{R}$ to form a knowledge graph representing information the agent has learned about the world. Examples of questions include: "What is my current location?", "What objects are around me?", and "What am I carrying?" to respectively extract information regarding the agent's current location, surrounding objects, inventory objects. Further, we predict attributes for each object by asking the question "What attributes does $x$ object have?". An example of the knowledge graph that can be extracted from description text and the overall Q*BERT architecture are shown in Figure 3.

For question-answering, we use the pre-trained language model, ALBERT (Lan et al., 2020), a variant of BERT that is fine-tuned for question answering on the SQuAD 2.0 (Rajpurkar et al., 2018) question-answering dataset. We further fine-tune the ALBERT model on a dataset specific to the text-game domain. This dataset, dubbed *Jericho-QA*, was created by making question answering pairs about text-games in the *Jericho* (Hausknecht et al., 2020)[1] framework as follows: For each game in Jericho, we use an oracle—an agent capable of playing the game perfectly using information normally off-limits such as the true game state—and a random exploration agent to gather ground truth state information about locations, objects, and attributes. From this ground truth, we construct pairs of questions in the form that Q*BERT will ask as it encounters environment description text, and the corresponding answers. These question-answer pairs are used to fine-tune the Q/A model and the ground truth data are discarded. No data from games we test Q*BERT on are used during ALBERT fine-tuning. Additional details can be found in Appendix A.1.

In a text-game the observation is a textual description of the environment. For every observation received, Q*BERT produces a fixed set of questions. The questions and the observation text are sent to the question-answering system. Predicted answers are converted into $\langle s, r, o \rangle$ triples and added to the knowledge graph. The complete knowledge graph is the input into Q*BERT's neural architecture (described below), which makes a prediction of the next action to take.

---

[1] https://github.com/microsoft/jericho

**Q*BERT Training** At every step an observation consisting of several components is received: $o_t = (o_{t_{desc}}, o_{t_{game}}, o_{t_{inv}}, a_{t-1})$ corresponding to the room description, game feedback, inventory, and previous action, and total score $R_t$. The room description $o_{t_{desc}}$ is a textual description of the agent's location, obtained by executing the command "look". The game feedback $o_{t_{game}}$ is the simulators response to the agent's previous action and consists of narrative and flavor text. The inventory $o_{t_{inv}}$ and previous action $a_{t-1}$ components inform the agent about the contents of its inventory and the last action taken respectively.

Each of these components is processed using a GRU based encoder utilizing the hidden state from the previous step and combined to have a single observation embedding $\mathbf{o}_t$. At each step, we update our knowledge graph $G_t$ using $o_t$ as described in earlier in Section 4 and it is then embedded into a single vector $\mathbf{g_t}$. This encoding is based on the R-GCN and is calculated as:

$$\mathbf{g_t} = f\left(\mathbf{W_g}\sigma\left(\sum_{r\in\mathcal{R}}\sum_{j\in\mathcal{N}_i^r}\frac{1}{c_{i,r}}\mathbf{W_r}^{(l)}\mathbf{h}_j^{(l)} + \mathbf{W_0}^{(l)}\mathbf{h}_i^{(l)}\right) + \mathbf{b_g}\right) \tag{2}$$

Where $\mathcal{R}$ is the set of relations, $\mathcal{N}_i^r$ is the 1-step neighborhood of a vertex $i$ with respect to relation $r$, $\mathbf{W_r}^{(l)}$ and $\mathbf{h}_j^{(l)}$ are the learnable convolutional filter weights with respect to relation $r$ and hidden state of a vertex $j$ in the last layer $l$ of the R-GCN respectively, $c_{i,r}$ is a normalization constant, and $\mathbf{W_g}$ and $\mathbf{b_g}$ the weights and biases of the output linear layer. The full architecture can be found in Fig. 3. The state representation consists only of the textual observations and knowledge graph. Another key use of the knowledge graph, introduced as part of KG-A2C, is the *graph mask*, which restricts the possible set of entities that can be predicted to fill into the action templates at every step to those found in the agent's knowledge graph. The rest of the training methodology is unchanged from Ammanabrolu & Hausknecht (2020), more details can be found in Appendix A.1.

## 5 STRUCTURED EXPLORATION

This section describes MC!Q*BERT an exploration method built on Q*BERT that detects overcomes bottlenecks by backtracking and policy chaining. This method of chaining policies and backtracking can be thought of in terms of *options* (Sutton et al., 1999; Stolle & Precup, 2002), where the agent decomposes the task of solving the text game into the sub-tasks, each of which has it's own policy. In our case, each sub-task delivers the agent to a bottleneck state.

### 5.1 BOTTLENECK DETECTION USING INTRINSIC MOTIVATION

Inspired by McGovern & Barto (2001), we present an intuitive way of detecting bottleneck states such as those in Fig. 2—or sub-tasks—in terms of whether or not the agent's ability to collect reward stagnates. If the agent does not collect a new reward for a number of environment interactions—defined in terms of a *patience* parameter—then it is possible that it is stuck due to a bottleneck state. An issue with this method, however, is that the placement of rewards does not always correspond to an agent being stuck. Complicating matters, rewards are sparse and often delayed; the agent not collecting a reward for a while might simply indicate that further exploration is required instead of truly being stuck.

To alleviate these issues, we define an *intrinsic motivation* for the agent that leverages the knowledge graph being built during exploration. The motivation is for the agent to learn more information regarding the world and expand the size of its knowledge graph. This provides us with a better indication of whether an agent is stuck or not—a stuck agent does not visit any new states, learns no new information about the world, and therefore does not expand its knowledge graph—leading to more effective bottleneck detection overall. To prevent the agent from discovering reward loops based on knowledge graph changes, we formally define this reward in terms of new information learned.

$$r_{\text{IM}_t} = \Delta(\mathcal{KG}_{\text{global}} - \mathcal{KG}_t) \text{ where } \mathcal{KG}_{\text{global}} = \bigcup_{i=1}^{t-1}\mathcal{KG}_i \tag{3}$$

Here $\mathcal{KG}_{\text{global}}$ is the set of all edges that the agent has ever had in its knowledge graph and the subtraction operator is a set difference. When the agent adds new edges to the graph perhaps as a

the result of finding a new room $\mathcal{KG}_{\text{global}}$ changes and a positive reward is generated—this does not happen when that room is rediscovered in subsequent episodes. This is then scaled by the game score so the intrinsic motivation does not drown out the actual quest rewards, the overall reward the agent receives at time step $t$ looks like this:

$$r_t = r_{g_t} + \alpha r_{\text{IM}_t} \frac{r_{g_t} + \epsilon}{r_{\max}} \tag{4}$$

where $\epsilon$ is a small smoothing factor, $\alpha$ is a scaling factor, $r_{g_t}$ is the game reward, $r_{\max}$ is the maximum score possible for that game, and $r_t$ is the reward received by the agent on time step $t$.

## 5.2 MODULAR POLICY CHAINING

A primary reason that agents fail to pass bottlenecks is not satisfying all the required dependencies. To solve this problem, we introduce a method of policy chaining, where the agent utilizes the determinism of the simulator to backtrack to previously visited states in order to fulfill dependencies required to overcome a bottleneck.

Specifically, Algorithm 1 optimizes the policy $\pi$ as usual, but also keeps track of a buffer $\mathcal{S}$ of the distinct states and knowledge graphs that led up to each state (we use state $s_t$ to colloquially refer to the combination of an observation $o_t$ and knowledge graph $\mathcal{KG}_t$). Similarly, a bottleneck buffer $\mathcal{S}_b$ and policy $\pi_b$ reflect the sequence of states and policy with the maximal return $\mathcal{J}_{\max}$—consisting of the cumulative intrinsic as well as game rewards. A bottleneck is identified when the agents fails to improve upon $\mathcal{J}_{\max}$ after *patience* number of steps, i.e. no improvement in raw game score or knowledge-graph-based intrinsic motivation reward. The agent then *backtracks* by searching back-

---

**Algorithm 1** Structured Exploration

$\{\pi_{\text{chain}}, \pi_b, \pi\} \leftarrow \phi$  ▷ Chained, backtrack, current policy
$\{\mathcal{S}_b, \mathcal{S}\} \leftarrow \phi$  ▷ Backtrack, current state buffers
$s_0, r_{\text{init}} \leftarrow \text{ENV.RESET}()$
$\mathcal{J}_{\max} \leftarrow r_{\text{init}}, p \leftarrow 0$
**for** timestep t in 0...M **do**  ▷ Train for M Steps
$\quad s_{t+1}, r_t, \pi \leftarrow \text{Q*BERTUPDATE}(s_t, \pi)$
$\quad \mathcal{S} \leftarrow \mathcal{S} + s_{t+1}$  ▷ Append current state to state buffer
$\quad p \leftarrow p + 1$  ▷ Lose patience
$\quad$ **if** $\mathcal{J}(\pi) \leq \mathcal{J}_{\max}$ **then**
$\quad\quad$ **if** $p \geq patience$ **then**  ▷ Stuck at a bottleneck
$\quad\quad\quad s_t, r_{\max}, \pi \leftarrow \text{BACKTRACK}(\pi_b, \mathcal{S}_b)$
$\quad\quad\quad$ ▷ Bottleneck passed; Add $\pi$ to the chained policy
$\quad\quad\quad \pi_{\text{chain}} \leftarrow \pi_{\text{chain}} + \pi$
$\quad$ **if** $\mathcal{J}(\pi) > \mathcal{J}_{\max}$ **then**  ▷ New highscore found
$\quad\quad \mathcal{J}_{\max} \leftarrow \mathcal{J}(\pi); \pi_b \leftarrow \pi; \mathcal{S}_b \leftarrow \mathcal{S}; p \leftarrow 0$
**return** $\pi_{\text{chain}}$  ▷ Chained policy that reaches max score

**function** Q*BERTUPDATE$(s_t, \pi)$  ▷ One-step update
$\quad s_{t+1}, r_{g_t} \leftarrow \text{ENV.STEP}(s_t, \pi)$  ▷ Section 4
$\quad r_t \leftarrow \text{CALCULATEREWARD}(s_{t+1}, r_{g_t})$  ▷ Eq. 4
$\quad \pi \leftarrow \text{A2C.UPDATE}(\pi, r_t)$  ▷ Appendix A.1
$\quad$ **return** $s_{t+1}, r_t, \pi$

**function** BACKTRACK$(\pi_b, \mathcal{S}_b)$  ▷ Try to overcome bottleneck
$\quad$ **for** b in REVERSE$(\mathcal{S}_b)$ **do**  ▷ States leading to highscore
$\quad\quad s_0 \leftarrow b; \pi \leftarrow \phi$
$\quad\quad$ **for** timestep $t$ in 0...N **do**  ▷ Train for N steps
$\quad\quad\quad s_{t+1}, r_t, \pi \leftarrow \text{Q*BERTUPDATE}(s_t, \pi)$
$\quad\quad\quad$ **if** $\mathcal{J}(\pi) > \mathcal{J}(\pi_b)$ **then return** $s_t, r_t, \pi$
$\quad$ **Terminate**  ▷ Can't find better score; Give up.

---

wards through the state sequence $\mathcal{S}_b$, restarting from each of the previous states—and training for $N$ steps in search of a more optimal policy to overcome the bottleneck. When such a policy is found, it is appended to modular policy chain $\pi_{\text{chain}}$. Conversely, if no such policy is found, then we have failed to pass the current bottleneck and the training terminates.

## 6 EVALUATION

We first evaluate the quality of the knowledge graph construction in a supervised setting. Next we perform an end-to-end evaluation in which knowledge graph construction is used by Q*BERT. We further measure the utility of the knowledge graph-based intrinsic motivation in bottleneck detection and conduct an empirical comparison between MC!Q*BERT and other exploration strategies.

### 6.1 GRAPH EXTRACTION EVALUATION

Table 1 (columns 2-5) shows the QA performance, and consequently the accuracy of the knowledge graphs built during exploration, on the Jericho-QA dataset using the rules-based approach of KG-A2C and the trained Albert-QA model in Q*BERT. Exact match (EM) is the percentage of times the model

| Expt. | QA Graph accuracy | | | | Game reward | | | | | Intrinsic | |
|---|---|---|---|---|---|---|---|---|---|---|---|
| Agent | KG-A2C | | Q*BERT | | KG-A2C | | Q*BERT | | MC!Q* | MC!Q* | GO!Q* |
| Metric | EM | F1 | EM | F1 | Eps. | Max | Eps. | Max | Max | Max | Max |
| zork1 | 6.08 | 8.42 | 43.93 | 48.31 | 34 | 35 | 34.1 | 35 | 32 | 41.6 | 31 |
| library | 10.33 | 26.74 | 49.78 | 52.76 | 14.3 | 19 | 10.0 | 18 | 19 | 19 | 18 |
| detective | 7.51 | 10.23 | 60.28 | 63.21 | 207.9 | 214 | 246.1 | 274 | 320 | 330 | 304 |
| balances | 32.53 | 36.09 | 85.81 | 86.18 | 10 | 10 | 10 | 10 | 10 | 10 | 10 |
| pentari | 16.48 | 23.36 | 65.02 | 69.54 | 50.7 | 56 | 51.2 | 56 | 56 | 58 | 40 |
| ztuu | 14.40 | 21.74 | 49.44 | 49.82 | 6 | 9 | 5 | 5 | 5 | 11.8 | 5 |
| ludicorp | 14.47 | 18.48 | 57.58 | 60.95 | 17.8 | 19 | 18 | 19 | 19 | 22.8 | 20.6 |
| deephome | 3.34 | 3.86 | 9.31 | 9.84 | 1 | 1 | 1 | 1 | 8 | 6 | 1 |
| temple | 7.42 | 9.44 | 48.98 | 49.17 | 7.6 | 8 | 7.9 | 8 | 8 | 8 | 8 |

Table 1: QA results (EM and F1) on Jericho-QA test set and averaged asymptotic scores on games by different methods across 5 independent runs. For KG-A2C and Q*BERT, we present scores averaged across the final 100 episodes as well as *max scores*. Methods using exploration strategies show only *max scores* because Episode Average Score (*Eps.*) conflates forward progress and backtracking. Agents are allowed $10^6$ steps for each parallel A2C agent with a batch size of 16.

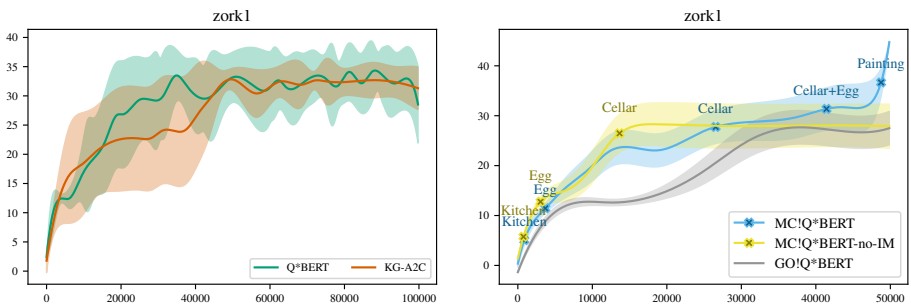

(a) Episode rewards for KG-A2C and Q*BERT. (b) Max reward curves for exploration strategies.

Figure 4: Select ablation results on *Zork1* conducted across 5 independent runs per experiment. We see where the agents using structured exploration pass each bottleneck seen in Fig. 2. Q*BERT without IM is unable to detect nor surpass bottlenecks beyond the *Cellar*.

was able to predict the exact answer string, while F1 measures token overlap between prediction and ground truth. We observe a direct correlation between the quality of the extracted graph and an agent's performance on the games—Q*BERT in general possessing knowledge graphs of much higher quality than KG-A2C. On games where Q*BERT performed comparatively better than KG-A2C in terms of asymptotic scores (columns 7 and 9), e.g. *detective*, the QA model had relatively high EM and F1, and vice versa as seen with *ztuu*. In general Q*BERT reaches comparable asymptotic performance to KG-A2C on 7 out of 9 games. However, as illustrated on *zork1* in Figure 4a, Q*BERT reaches asymptotic performance faster than KG-A2C, indicating that the QA model improves learning; this trend is consistent on other games as shown in additional plots in Appendix B. Both agents rely on the graph to constrain the action space and provide a richer input state representation. Q*BERT uses a QA model fine-tuned on regularities of a text-game producing more relevant knowledge graphs than those extracted by OpenIE (Angeli et al., 2015) in KG-A2C for this purpose.

## 6.2 INTRINSIC MOTIVATION AND EXPLORATION STRATEGY EVALUATION

We evaluate intrinsic motivation through policy chaining, dubbed **MC!Q*BERT** (Modularly Chained Q*BERT) by first testing policy chaining with only game reward and then with both game reward and intrinsic motivation. We provide a qualitative analysis of the bottlenecks detected with both methods with respect to those found in Fig. 2 on *Zork1*. Because MC!Q*BERT exploits structural domain assumptions that Q*BERT and KG-A2C cannot, we create a strong alternative baseline that looks at whether modular chaining improves over a related exploration strategy used in Go-Explore (Ecoffet et al., 2019). **GO!Q*BERT** is a baseline that makes the same underlying assumptions regarding the simulator as MC!Q*BERT but operates differently by tracking sub-optimal and under-explored states in order to allow the agent to explore upon more optimal states that may be a result of sparse rewards. This baseline trains Q*BERT in parallel to generate actions from the full action space used for

exploration. Further details are found in Appendix A.3. When MC!Q*BERT only uses game reward it matches Q*BERT on 5 out of 9 games and outperforms on 3 out of 9 games. When MC!Q*BERT uses intrinsic motivation plus game reward, it strictly outperforms KG-A2C and Q*BERT on 6 out of 9 games and matches it on the rest. MC!Q*BERT outperforms GO!Q*BERT on 7 games and matches on 2, indicating that the modular chaining exploration strategy exploits the intrinsic motivation of knowledge graph learning better than the closest alternative exploration strategy.

## 7 ANALYSIS

Table 1 shows that across all the games MC!Q*BERT matches or outperforms the current state-of-the-art when compared across the metric of the max score consistently received across runs. There are two main trends: First, MC!Q*BERT strongly benefits from the inclusion of intrinsic motivation rewards. Qualitatively, we illustrate this with *Zork1*, the canonical commercial text-adventure game that no RL agent has ever beaten. An analysis of bottlenecks detected by each agent in this game reveals differences in the overall accuracy of the bottleneck detection between MC!Q*BERT with and without intrinsic motivation. With intrinsic motivation, across 5 independent runs, MC!Q*BERT had an average true positive bottleneck state detection rate of 63%, false positive of 37%, with 50% coverage; and without it has a true positive rate of 58%, false positive of 42%, with coverage of 25%—assuming that the states such as in Fig. 2 represent the ground truth for bottlenecks. Coverage here refers to the number of unique bottlenecks states found during exploration compared to the total number of such states in the ground truth. This indicates that overall quality of bottleneck detection significantly improves given intrinsic motivation—enabling MC!Q*BERT to backtrack and surpass them. Figure 4b shows when each of these agents detect and subsequently overcome the bottlenecks outlined in Figure 2.

When intrinsic motivation is not used, the agent discovers that it can get to the *Kitchen* with a score of $+10$ and then *Cellar* with a score of $+25$ immediately after. It forgets how to get the *Egg* with a smaller score of $+5$ and never makes it past the Grue in the *Cellar*. Intrinsic motivation avoids this in two ways: (1) it makes it less focused on a locally high-reward trajectory—making it less greedy and helping it chain together rewards for the *Egg* and *Cellar*, and (2) provides rewards for fulfilling dependencies that would otherwise not be rewarded by the game—this is seen by the fact that it learns that picking up the lamp is the right way to surpass the *Cellar* bottleneck and reach the *Painting*. A similar behavior is observed with GO!Q*BERT: the agent settles prematurely on a locally high-reward trajectory and thus never has incentive to find more globally optimal trajectories by fulfilling the underlying dependency graph. Here, the likely cause is due to GO!Q*BERT's inability to backtrack and rethink discovered locally-maximal reward trajectories.

The second trend is that using both the improvements to graph construction in addition to intrinsic motivation and structured exploration consistently yields higher max scores across a majority of the games when compared to the rest of the methods. Having just the improvements to graph building or structured exploration by themselves is not enough. Thus we infer that the full MC!Q*BERT agent is fundamentally exploring this combinatorially-sized space more effectively by virtue of being able to more consistently detect and clear bottlenecks. The improvement over systems using default exploration such as KG-A2C or Q*BERT by itself indicates that structured exploration is necessary when dealing with sparse and ill-placed reward functions.

## 8 CONCLUSIONS

Modern deep reinforcement learning agents using default exploration strategies such as $\epsilon$-greedy are ill-equipped to deal with the latent structure of dependencies and bottlenecks found in many text-based games. To help address this challenge, we introduced two new agents: Q*BERT, an agent that constructs a knowledge graph of the world by asking questions about it, and MC!Q*BERT, which uses intrinsic motivation to grow the graph and detect bottlenecks arising from delayed rewards. A key insight from ablation studies is that the graph-based intrinsic motivation is crucial for bottleneck detection, preventing the agent from falling into locally optimal high reward trajectories due to ill-placed rewards. Policy chaining used in tandem with intrinsic motivation results in agents that explore further in the game by clearing bottlenecks more consistently.

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
