# OpenReview forum: "How to Avoid Being Eaten by a Grue: Structured Exploration Strategies for Textual Worlds"
_ICLR.cc/2021/Conference — Reject_

### Official Review · AnonReviewer1 · 2020-10-28
**Interesting Application with Large Number of New Methods**

**Rating:** 6
**Confidence:** 2

**Review:**

Summary/Overall Quality: The authors make the following contributions.
- focus on text based adventure games
- create agent that can build knowledge graph by answering question
- introduce novel exploration strategy
- IM reward: expand size of knowledge graph
Because the authors present a number of interesting and well tested strategies for a well-justified task, the paper is above the acceptance threshold. If the clarity could be improved (specifically by making the contribution in the general case more explicit) and if the experiments could be made more thorough (+ a stronger improvement of prior work) it would be a strong paper.

Clarity: The authors did a reasonable job of familiarizing readers with the challenges of text-based games. However, the description of the research contribution and its potential impacts could have been more focused. For example, in the abstract the authors don't mention that they collect a Jericho-QA dataset, but it is listed as one of the three contributions in the final paragraph of the related work. The paper spans a lot of material--from the research merit of text based games to intrinsic motivation based on knowledge graph formation to using QA models for knowledge graph construction to modular policy chaining--and it is difficult to walk away from the paper with a cohesive understanding of the research contribution.

Originality: While the individual components are not extremely novel, taken together they create an interesting and original system design. E.g., even though the notion of using intrinsic motivation to un-sparsify a reward space is far from new, the authors devise an interesting formulation of IM to apply to knowledge graph construction.

Significance:
- _For text based games_: the authors provide numerous method contributions and a dataset for this application specifically. However, looking at table 1 and figure 4, its not clear that the fullest version of their method has a consistent advantage over prior work.
- _Generally_ Text-based games could be an important benchmark/milestone for systems that need to reason well into the past and over a large action, discrete action space. The authors do not address this much beyond some prose in the abstract and introduction. Certainly some components of the methods they have developed (e.g., exploration over a knowledge graph) has broad-spanning applications.

Strengths:
- The authors provide a variety of methods to help agents excel is text based games: knowledge graph creation with a QA system, a dataset to train the QA system, intrinsic motivation based on knowledge graph expansion, modular policy training.
- Show results over a large number of text-based games. On several games, their model, MC!Q* sees a substantial improvement over prior work.

Weaknesses:
- Improvement over prior work is not consistent (see library, balances, and temple tasks in table 1, in figure 4a Q*bert converges more quickly on average but not above variance of KG-A2C)
- As someone unfamiliar with text based games, it's hard to interpret the results in table 2 and to understand what problems are being addressed in each game.
- More experiments to better understand how bottlenecks are addressed by the model (e.g., more of figure 4b) would be enlightening

---

> ### Author Response · Authors · 2020-11-15
> **Addressing some weaknesses**
>
> We thank the reviewer for their time and effort and are encouraged by their analysis of this work's significance. We will attempt to address a couple of the weaknesses pointed out.
>
> 1. Improvement over prior work is not consistent (see library, balances, and temple tasks in table 1, in figure 4a Q\*bert converges more quickly on average but not above variance of KG-A2C)
> - Yes - as mentioned Q\*BERT does not systematically improve upon asymptotic scores of previous work. While the BERT-QA knowledge graph creation routine leads to more accurate knowledge graphs, using more accurate knowledge graphs alone did not lead to improvements in asymptotic performance. This finding is interesting because it suggests that other improvements may be needed in the agent in order to fully leverage improvements made to the knowledge graphs. One such agent improvement was MC!Q\*BERT which uses the improved knowledge graph accuracy as a way to accurately detect and overcome bottlenecks. However, we believe that in future work there may be many other ways for agents to utilize more accurate knowledge graphs such as those constructed by Q\*BERT.
>
> 2. As someone unfamiliar with text based games, it's hard to interpret the results in table 2 and to understand what problems are being addressed in each game.
> - Yes, it’s certainly difficult to relate changes in numerical score to the more qualitative aspects of improvement on each game. While there’s no true replacement for being familiar with each game, we will seek to include more transcripts and game logs produced by the agents to get a better sense of the challenges and bottlenecks presented by the different games.

---

### Official Review · AnonReviewer3 · 2020-10-28
**interesting idea; limited novelty; misleading claims; unclear presentation; reject**

**Rating:** 4
**Confidence:** 4

**Review:**


The paper proposes two RL agents for the text adventure games: QBERT learns (relational) knowledge about the game world with assistance of a trained QA-model and then constrains its action space with help of the knowledge base, enjoying greater sample efficiency; MCQBERT learns to escape from local optima by backtracking through the navigation trajectories, guided by the learned knowledge base and a manually-designed intrinsic reward function.

Pros:

This is a very dense paper with multiple key ideas, namely (1) using QA model to construct knowledge base; (2) escaping local optima by backtracking; and (3) modular policy chaining.

Both (1) and (2) are interesting and novel. But the novelty of (1) is limited because using QA to build graph has been used by e.g., Ammanabrolu et al. Bringing Stories Alive: Generating Interactive Fiction Worlds.

Empirical results demonstrate the effectiveness of the proposed methods. Not being eaten by Grue is impressive.

Cons:

Presentation is the main reason that I’d like to reject the current version:
(1) Several key claims are misleading and over-bold.
(2) Many essential technical details are not clear.

[Misleading/Over-bold claims]

The authors claimed, at many places, that they learned the "dependencies'', namely the quest structure. For example, they said the agent "learns that picking up the lamp is the right way to surpass the Cellar bottleneck and reach the Painting".

This claim is wrong: the agent only successfully passes that room because it has done enough exploration (and the exploration is efficient because the intrinsic reward function is cleverly designed).

If this comment is not clear enough, let me be a bit more specific.
In the "lamp-Grue" example, the agent doesn't know the sense like "if I pick up this lamp, I can pass the Grue room'' or "now I am in the Grue room and the lamp is helpful".
Instead, the agent only picks the lamp up because it is encouraged to discover new information.

Or, in other words, the agent doesn't really learn the quest structure as shown in Figure-2.
(But I agree that Figure-2 is useful to illustrate the interesting idea.)

I believe that the right way to view the method is: exploration is guided by not only environment reward but also intrinsic desire to discover new information, which leads to more efficient exploration and higher long-term reward.
Similar problems in other RL domains have been tackled and here is a paper that is similar to this submission in spirit:
Conti et al NeurIPS 2018 Improving Exploration in Evolution Strategies for Deep Reinforcement Learning via a Population of Novelty-Seeking Agents.
Quote them: "reward functions are often deceptive, and solely optimizing for reward without some mechanism to encourage intelligent exploration can lead to getting stuck in local optima"---same problem as in text adventure games.
In this submission, the agents also rely on the learned knowledge base.
Note: the learned knowledge base is very different from the quest (or dependency) structure---the former is like Figure-3 while the latter is Figure-2.

Then in what sense can the authors confidently claim that their agents learned things like "the virtue of being able to more consistently detect and clear bottlenecks" or "deal with latent structure of dependencies and bottlenecks"? My opinion is: only if they can actually learn some structures like shown in Figure-2.

That being said, I request the authors to rewrite all the claims like mentioned above.

[Technical clarity]

The most important clarity problem is about the modular policy chaining method: it is not clear how the method is integrated with the entire framework.
E.g., how is $\pi_{\text{chain}}$ is used? It is only mentioned in Algo-1 but nowhere else.
Other related detailed questions include:
What is the "policy" in this paper? Is it a function to be learned, or a set of parameters, or a specific action for a given state? When a policy is buffered or chained, what is actually stored and how?
Algorithm-1 always has "Bottleneck passed". What if the bottleneck isn't passed? BACKTRACK procedure may not find a better score, so bottleneck is not passed, is it consistent with "bottleneck passed"?
Is the BACKTRACK algorithm greedy? It looks like it returns immediately once it finds a higher $\mathcal{J}$?
How is $\mathcal{J} computed?$ It looks like it is stored and then indexed somehow. But how? How can you index a value by a "policy" as a key? This is related to the ``what is policy" question.

There are other things about the framework that need to be clarified as well.

Is the training algorithm online or offline? In Algo-1, A2C is called, indicating it is trained (step by step) during exploration: thus it seems an online algorithm. But what about QBERT which doesn't have modular policy chaining?

What is the full action space? There should be some discussion about it in main paper, referring to the related appendices then.

How could one know the "maximum score possible for the game"?

Does the agent ask the same set of questions (to QA model) every step? Then how could it handle consistency? E.g., if "what am I carrying" is asked many times while your inventory is not changed at all, would it generate exactly the same answer all the time and how could you handle any difference?

In Figure-3, why ALBERT-QA doesn’t point to $KG$ but only points to $V_t$? Is ALBERT-QA used specifically to build knowledge graph?
In general, it is a little hard to connect the elements of Figure-3 to formula in section-4.

Here are some presentation issues about the math formula:
Index-$i$ and index-$j$ in eqn-(1) are not grounded: they are mentioned in text; they don't loop over anything.
Similar problem for index-$i$ in eqn-(2).

Here are some minor presentation issues:
"MCQBERT an agent that uses" -> "MC!QBERT, an agent that uses"
"freedom to a explore" -> "freedom to explore"
"these dependency graphs are … either … or … to progress and … a priori" -> this sentence is too convoluted to correctly parse
"mostly deterministic … probabilities" -> odd phrase: do you mean degenerative distribution?
"and without it has a true positive …" -> ungrammatical

---

> ### Author Response · Authors · 2020-11-15
> **Clarifying presentation and claims**
>
> We would first like to thank the reviewer for their thoughtful comments and time. We will now address some of the expressed concerns below.
>
> 1. Re: The authors claimed, at many places, that they learned the "dependencies'', namely the quest structure. For example, they said the agent "learns that picking up the lamp is the right way to surpass the Cellar bottleneck and reach the Painting". ... That being said, I request the authors to rewrite all the claims like mentioned above.
> - Thanks for this feedback. We agree that MC!Q*BERT is not explicitly learning or reasoning about the latent dependency structure of the game. We contend that MC!Q*BERT does learn to detect bottlenecks (such as the darkness in the Cellar) and attempts to overcome them via strategic exploration guided by knowledge-graph expansion. We further agree that the agent is not learning the quest structure as shown in Figure-2, but also contend that by simply detecting and overcoming bottlenecks, we can nevertheless make progress in games exhibiting this type latent structure. To your point, we will remove any claims that imply MC!Q*BERT is learning dependencies or quest structure.
>
> 2. Re: The most important clarity problem is about the modular policy chaining method: it is not clear how the method is integrated with the entire framework. ... (And associated questions)
> - $\pi_{chain}$ is mentioned in the prose of Section 5.2 as being the overall modular policy to which each individual bottleneck beating policy is appended to. The policy refers to all three of the things mentioned but implementation wise refers to the parameters of the architecture seen in Fig. 3, which outputs an action and a value given an input state. These parameters are what is stored. If a bottleneck isn’t passed the training terminates (last line of Algo 1). Similarly, BACKTRACK by itself is greedy in that it returns as soon as a higher $\mathcal{J}$ is found. $\mathcal{J}$ is computed by calculating the score that would be achieved if that policy is executed. The indexing happens the other way around, i.e. policies (parameters for the policy network) are indexed by the value $\mathcal{J}$.
>
> 3. Is the training algorithm online or offline? In Algo-1, A2C is called, indicating it is trained (step by step) during exploration: thus it seems an online algorithm. But what about QBERT which doesn't have modular policy chaining?
> - All algorithms presented are online. Q*BERT is also trained using just vanilla A2C as KG-A2C is.
>
> 4. What is the full action space? There should be some discussion about it in main paper, referring to the related appendices then.
> - The full action space is unchanged from KG-A2C and so is mentioned only in the appendices. We use the full template space as proposed in Hausknecht at al. 2020 in the original Jericho work. This consists of n templates per game, with a max of 2 possible object slots that can be filled in with a vocabulary of size m. The values of m, n can be found in the original Jericho work.
>
> 5. How could one know the "maximum score possible for the game"?
> - These are defined as part of the Jericho benchmark.
>
> 6. Does the agent ask the same set of questions (to QA model) every step? Then how could it handle consistency? E.g., if "what am I carrying" is asked many times while your inventory is not changed at all, would it generate exactly the same answer all the time and how could you handle any difference?
> - All questions possible are given to the agent at every step but the Jericho-QA data is formatted in the style of SQuAD 2.0 and given samples of which questions are not applicable to certain states and also pretrained on SQuAD 2.0. Thus, it is able to determine when to abstain from answering a question and will output nothing if it is "unsure". In cases when the inventory is unchanged, it outputs the same answer anytime. If a difference is found the newest answer overrides previous answers.  This is mentioned in Appendix A.1.
>
> 7. In Figure-3, why ALBERT-QA doesn’t point to KG but only points to V_t? Is ALBERT-QA used specifically to build knowledge graph? In general, it is a little hard to connect the elements of Figure-3 to formula in section-4.
> - ALBERT-QA is used to extract information regarding the surroundings via the questions resulting a typed vertex set. This is then combined with the previous graph using rules as mentioned in the main paper (the exact rules are found in Appendix A.1). Hence, ALBERT-QA points only to V_t and not KG_t.

---

### Official Review · AnonReviewer2 · 2020-10-28
**Interesting paper, with some minor issues, but significantly contributing to the state of the art.**

**Rating:** 7
**Confidence:** 4

**Review:**

Summary:

This paper focuses on learning to play text adventure games using reinforcement learning. The paper presents two new algorithms: Q*BERT and MC!Q*BERT, as well as a QA dataset to help training a component of these agents that builds a knowledge graph of the current game state based on the textual descriptions received by from the game.

Reasons for score:

I gave this paper a 7, as I think the overall contributions to the text-based adventure game playing literature are strong. There are a few minor technical issues here and there (see my detailed feedback below), but those are minor things that can be easily fixed. In particular, the new approach to integrate exploration strategies to overcome "bottlenecks" in the search space is interesting and, to the best of my knowledge, novel.

Additional feedback:

- page 1: "We contend that existing Reinforcement Learning agents that are unaware of" -> "We contend that existing Reinforcement Learning agents are unaware of"
- page 2: nit, "In text-adventure games the dependencies are of two types" -> I can recall many bottlenecks from text adventure games that do not fit either of these two categories. So, perhaps instead of "are of two types" you could say that these are the two dependencies you considered (e.g., some games require many unrewarded actions (in terms of in-game points) like visiting locations more than once, or visiting them at some particular times, or even interacting with certain characters in certain ways, etc.). Basically, location and inventory are not the only state of the game, but there are many other state variables in many of these games.
- I am not convinced that the dependency graph of Zork can be divided into a linear set of  "level" subdivisions the authors state before Equation 1. For example, how about branches? If there are parallel tasks that need to be performed each with their own "bottlenecks" (for example, completing three tasks that can be done in any order), then the "j>i" condition in Equation 1 breaks, as it will result in some spurious bottlenecks. Thus, I think this part of the paper needs some work (authors mention "relatively linear plots", but "relative linear" does not mean "completely linear"). Edit after reaching Section 5: and since this definition is not the one used by MC!Q*BERT anyway, why have it in the paper in any case?
- Table 1: bolding the highest scores for each game would be useful to understand the table at a glance.
- page 8: about the bottleneck identification rates reported: how were ground truths established? was this manually labeled? If it was automatically labeled, I'm not sure if the definition used earlier in the paper was used, but if it was, I am not confident it is a good definition.

---

> ### Author Response · Authors · 2020-11-15
> **Addressing minor issues**
>
> We thank the reviewer for their encouraging and helpful comments.
>
> 1. I am not convinced that the dependency graph of Zork can be divided into a linear set of "level" subdivisions the authors state before Equation 1. For example, how about branches? If there are parallel tasks that need to be performed each with their own "bottlenecks" (for example, completing three tasks that can be done in any order), then the "j>i" condition in Equation 1 breaks, as it will result in some spurious bottlenecks. Thus, I think this part of the paper needs some work (authors mention "relatively linear plots", but "relative linear" does not mean "completely linear"). Edit after reaching Section 5: and since this definition is not the one used by MC!Q*BERT anyway, why have it in the paper in any case?
> - The dependency graph of Zork and introductions of level sets were intended to help build intuition about the structure of many text-based games. There is a distinction to be made between the dependency graph and the branching of the plot or the branching of the game itself. The dependency graph itself outlines the conditions an agent needs to meet in order to progress in the game and is always of the form given in the figure, even in cases where there are multiple parallel tasks (parallel tasks can be completed in any order will be expressed as being on the same topologically sorted level of the dependency graph, with j = i).
> - As you noted MC!Q*BERT does not explicitly construct or reason about dependency graphs but instead relies on the expansion of its knowledge graph to detect bottlenecks. Thanks for this suggestion, we will try to remove emphasis on the particular form of Equation 1.
> - As a note, the ground truths for bottleneck detection rates were manually labeled. We also incorporate the other page by page feedback given.

---

### Official Review · AnonReviewer4 · 2020-10-29
**Interesting approach, but have questions regarding experiment setup**

**Rating:** 5
**Confidence:** 4

**Review:**

This paper studies reinforcement learning setting where the agent's decision is augmented with external knowledge representation. Termed Q*BERT, this proposed method uses question answering to build a knowledge graph of the world. Results show the agent was able to pass the bottleneck for a popular game where most other algorithms failed.

Pros:
The proposed method seems well motivated and reasonable to improve agent's performance. Authors provide good review of text games' literature. Experimental results seem solid.

Cons:
1. The novelty in terms of methodology seems a bit low. The idea of Q*BERT training is not new.
2. I have some questions regarding authors' "Knowledge Graph State Representation". First of all, pretrained language models such as BERT contain rich information about world knowledge. For example, BERT will give a strong association between "door" and "key", while an agent that learns from scratch will not understand unless it receives an reward from "use key to open the door" (or the game hints the agent to do so). It's unclear why the authors argue using QA to augment agent's state representation. It is possible that using BERT's representation alone, the agent's performance will be much better and training will be more efficient already.
3. Another question relates to the way authors conduct this QA. The authors use an oracle agent to explore the word and a random agent to gather information such as attributes. Is there already information leakage during this process? In a fair experimental setup, there is no way an agent can foresee the world. Maybe I am misunderstanding here, and I hope the authors can help clarify.

---

> ### Author Response · Authors · 2020-11-15
> **Experimental setup clarifications**
>
> We thank the reviewer for their effort towards improving our manuscript and will make some clarifications below.
>
> 1. The novelty in terms of methodology seems a bit low. The idea of Q\*BERT training is not new.
> - Q\*BERT is trained with the same methodology as KG-A2C, but its novelty comes in the form of building a knowledge graph during exploration using question answering. As far as we are aware, this is the first method that uses this form of knowledge graph construction.
>
> 2. I have some questions regarding authors' "Knowledge Graph State Representation". First of all, pretrained language models such as BERT contain rich information about world knowledge. For example, BERT will give a strong association between "door" and "key", while an agent that learns from scratch will not understand unless it receives an reward from "use key to open the door" (or the game hints the agent to do so). It's unclear why the authors argue using QA to augment agent's state representation. It is possible that using BERT's representation alone, the agent's performance will be much better and training will be more efficient already.
> - The reviewer makes a good point, BERT certainly will give a strong association between those two phrases and leveraging these associations could be an interesting direction for future work. However, in this work we focus on using BERT for the purpose of constructing knowledge graphs because knowledge graphs have been shown to be advantageous in terms of providing a persistent memory to combat partial observability (Ammanabrolu et al. https://www.aclweb.org/anthology/N19-1358/) and for zero-shot generalization (Adhikari et al. https://arxiv.org/abs/2002.09127). MC!Q\*BERT further defines an effective way of detecting bottlenecks using knowledge graph building, which would not be possible solely using BERT embeddings.
>
> 3. Another question relates to the way authors conduct this QA. The authors use an oracle agent to explore the word and a random agent to gather information such as attributes. Is there already information leakage during this process? In a fair experimental setup, there is no way an agent can foresee the world. Maybe I am misunderstanding here, and I hope the authors can help clarify.
> - Attributes collected by the oracle agent, as well as the training is done on an entirely different set of games than the ones presented in this paper. This means the agent did not have any sort of oracle information for any of the games presented in this paper. We will clarify this in the appendix of future drafts.

---

### Decision · Program_Chairs · 2021-01-07
**Final Decision**

**Decision:**

Reject

**Comment:**

The paper describes very interesting work that advances the state of the art in Zork by going beyond an important state bottleneck.  While there is an important engineering contribution, the reviewers raised important concerns regarding the novelty of the question-answering approach to construct knowledge graphs, the clarity of the backtracking heuristic and the extent to which the proposed approach outperforms previous work.  I also read the paper and I agree with the concerns of the reviewers.  In particular, I encourage the authors to provide more details about the backtracking procedure, a formal description of the algorithm and its assumptions to help readers apply the approach in other domains, as well as a formal analysis to better understand when it will or will not pass a bottleneck state.